# Stable Enzymatic Nanoparticles from Nucleases, Proteases, Lipase and Antioxidant Proteins with Substrate-Binding and Catalytic Properties

**DOI:** 10.3390/ijms24033043

**Published:** 2023-02-03

**Authors:** Olga V. Morozova, Nikolay A. Barinov, Dmitry V. Klinov

**Affiliations:** 1Federal Research and Clinical Center of Physical-Chemical Medicine of Federal Medical Biological Agency, 1a Malaya Pirogovskaya Street, 119435 Moscow, Russia; 2Ivanovsky Institute of Virology of the National Research Center of Epidemiology and Microbiology Named after N.F. Gamaleya of the Russian Ministry of Health, 16 Gamaleya Street, 123098 Moscow, Russia; 3Moscow Institute of Physics and Technology, 9 Institutsky Per., 141700 Dolgoprudny, Russia

**Keywords:** DNase I, RNase A, trypsin, chymotrypsin, catalase, horseradish peroxidase, lipase, nanoprecipitation of proteins, enzymatic nanoparticles, broad size ranges, stability of the enzymes and aggregation of nanoparticles during storage, low cytotoxicity of bioactive enzymatic nanoparticles

## Abstract

Limited membrane permeability and biodegradation hamper the intracellular delivery of the free natural or recombinant enzymes necessary for compensatory therapy. Nanoparticles (NP) provide relative protein stability and unspecific endocytosis-mediated cellular uptake. Our objective was the fabrication of NP from 7 biomedicine-relevant enzymes, including DNase I, RNase A, trypsin, chymotrypsin, catalase, horseradish peroxidase (HRP) and lipase, the analysis of their conformation stability and enzymatic activity as well as possible toxicity for eukaryotic cells. The enzymes were dissolved in fluoroalcohol and mixed with 40% ethanol as an anti-solvent with subsequent alcohol evaporation at high temperature and low pressure. The shapes and sizes of NP were determined by scanning electron microscopy (SEM), atomic force microscopy (AFM) and dynamic light scattering (DLS). Enzyme conformations in solutions and in NP were compared using circular dichroism (CD) spectroscopy. The activity of the enzymes was assayed with specific substrates. The cytotoxicity of the enzymatic NP (ENP) was studied by microscopic observations and by using an MTT test. Water-insoluble ENP of different shapes and sizes in a range 50–300 nm consisting of 7 enzymes remained stable for 1 year at +4 °C without any cross-linking. CD spectroscopy of the ENP permitted us to reveal changes in proportions of α-helixes, β-turns and random coils in comparison with fresh enzyme solutions in water. Despite the minor conformation changes of the proteins in the ENP, the enzymes retained their substrate-binding and catalytic properties. Among the studied bioactive ENP, only DNase NP were highly toxic for 3 cell lines with granulation in 1 day posttreatment, whereas other NP were less toxic (if any). Taken together, the enzymes in the stable ENP retained their catalytic activity and might be used for intracellular delivery.

## 1. Introduction

Enzymes are biocatalysts with high substrate specificity, sensitivity and exceptional catalytic activity under mild physiological conditions [1]. They are used in many industrial and pharmaceutical areas for both diagnostics and compensation therapy of inherited and metabolic diseases [1,2,3,4]. Currently, exogenous enzymes are intravenously infused into the bloodstream of patients with enzyme deficiency. Mandatory frequent administration of enzymes leads to high costs and negatively impacts patient compliance [1,2,3]. Moreover, proteolytic hydrolysis, oxidation and partial denaturation in the presence of ligands and on surfaces may lower the performance of enzyme-based analytical devices or limit industrial efficiency, thus hampering the further implementation of natural enzymes [1,2,3,4,5]. Poor membrane permeability and biological barrier-crossing capacities of free enzymes constrain the treatment of diseases [1,2,3,4].

Great expectations of nanomedicine are caused by enhanced pharmacokinetic stability of NP, reduced toxicity, improved bioavailability and biodistribution. Successful enzyme delivery systems must be able to overcome several physiological barriers and release bioactive enzymes at the target organs or tissues, often inside the living cells. Currently, there are many types of nanocarriers of bioactive enzymes and enzyme-responsive drug delivery systems capable of releasing drugs in a controlled manner [2,3,4]. Liposomes, polymeric and inorganic NP can deliver encapsulated drugs on demand [2,3]. To increase stability, protein or medium engineering and chemical modification were used [4]. Immobilization of enzymes onto carriers, or entrapment within a matrix, framework or NP can also delay their degradation [1,4], but partial denaturation of proteins after attachment to solid support [5], aggregation or agglomeration cannot be excluded. Multiple enzyme molecules can be contained within nanomaterials or matrices of different origins. ENP consisting of glucose oxidase, HRP, uricase, cholesterol oxidase and hemoglobin were immobilized onto a matrix (electrode or membrane) in order to construct diagnostic biosensors [6]. A cell membrane-coated porous metal–organic zeolitic imidazolate framework has been suggested to encapsulate recombinant enzyme–uricase for intracellular delivery [7]. However, only 38% of the initial activity of the inputted uricase was retained in the metal–organic framework [7]. Moreover, an induced immune response may destroy foreign nanomaterials. The same concerns can be relevant to metallic or organic NP loaded with proteins or drugs [8,9,10].

ENP are the stable water-insoluble aggregates of enzymes or NP with surface enzymes of 10–500 nm in sizes capable of (1) maintaining enzymatic activity; (2) ensuring long-term stability against temperature, dehydration, organic solvents, and aggressive pH; and (3) enabling a tuning or reversible switching of enzyme activity. Enzymes can be stabilized by thin shells consisting of polymers prepared either by *in situ* polymerization or by assembling preformed polymers [4].

ENP provides many benefits, including increased enzyme stability, decreased degradation during circulation and targeting to the desired locations. Moreover, nanocarriers have the advantage of accumulating preferentially in solid tumors through the enhanced penetration and retention (EPR) effect. Other tumor-associated features include altered gene expression of enzymes and leaky vasculature [2]. Moreover, enzyme-containing nanostructures may be used for enzyme prodrug therapy for cancer. In particular, the enzyme HRP (which oxidizes the plant growth phytohormone—prodrug indole-3-acetic acid (IAA) to release toxic oxidative species) can be conjugated with Au NP for intracellular delivery in breast cancer cells. The nanodevice is biocompatible and effectively internalized by breast cancer cell lines. Co-treatment with HRP-Au NP and IAA reduced the viability of breast cancer cells below 5%, whereas the free formulated components (HRP, IAA) had no effect [11]. Overall, the ability of nanotechnology to improve the pharmacologic profile of a drug promises to increase efficacy while decreasing unspecific side effects.

Besides inorganic and organic NPs, virus-like particles (VLP) can be decorated with different drugs and enzymes. The attachment of the α-glucosidase Ima1p of *Saccharomyces cerevisiae* to the surface of parvovirus B19 VLP resulted in increased V_max_ toward p-nitrophenyl-α-D-glucopyranoside, a notorious 10 °C shift in its optimal temperature, from 35 °C for the non-attached enzyme to 45 °C for the enzyme attached to the NP. The decorated VLP may be further developed as part of therapy for the treatment of lysosomal storage diseases derived from defects in the human acid α-glucosidase [12].

A deoxyribonuclease (DNase) is a generic term for enzymes that catalyze the hydrolysis of phosphodiester bonds in the DNA. There are two main types of DNases found in metazoans: deoxyribonuclease I (*EC* 3.1.21.1) and deoxyribonuclease II. In biotechnology, DNase is used to degrade DNA from highly viscous bacterial lysates for the isolation of recombinant proteins. In medicine, DNase combined with intrapleural tissue plasminogen activator can be used to increase pleural drainage, decrease hospital length of stay, and decrease the need for surgery in parapneumonic effusions and empyema. DNase may also be used together with mucolytic agents like N-acetyl cysteine (NAC) to lower the viscosity of pulmonary mucosal secretion. They break the disulfide bonds linking the mucin monomers and depolarize the mucin network [13]. DNases derived from pathogens are necessary for wound infection monitoring.

Ribonuclease (RNase) is a type of nuclease that catalyzes the hydrolytic cleavage of RNA into shorter fragments. Ribonucleases can be divided into endoribonucleases and exoribonucleases, and comprise several sub-classes within the EC 2.7 (for the phosphorolytic enzymes) and 3.1 (for the hydrolytic enzymes) classes of enzymes. All known organisms contain many RNases of both classes, showing that RNA degradation is an evolutionary ancient and important process. RNases are required for the maturation of all types of RNA and their clearance. In addition, RNA degradation systems are the first defense lines against RNA-containing viruses and RNA interference. RNase A (EC 3.1.27.5) is specific to single-stranded RNAs. It cleaves the 3′-end of unpaired C and U residues, ultimately forming a 3′-phosphorylated product via a 2′,3′-cyclic monophosphate intermediate. RNase A from bovine pancreas is widely used in biomedical research and genetic engineering due to the developed methods of isolation and amazing stability. RNase 7, belonging to the RNase A superfamily is secreted by human skin and serves as a potent antipathogen defense. RNases take part in the unspecific protection of the respiratory tract due to antiviral and bactericidal activity.

Serine protease trypsin (EC 3.4.21.4) is abundant in pancreases and can be easily purified. Trypsin cuts peptide chains mainly on the carboxyl side of the amino acids lysine or arginine. Due to the developed procedure of isolation and known biochemical properties, trypsin is widely used in biotechnological processes (for resuspension and harvesting adhesive cells; for destruction of casein in breast milk to produce hypoallergenic food; in the extraction and control aroma formation in cheese and milk products; to tenderize meat) and medicine (in proteomics to digest proteins before mass spectrometry; to dissolve blood clots in its recombinant form and to treat inflammation in its pancreatic form). In veterinary, trypsin is an ingredient in wound spray to dissolve dead tissue and pus in wounds. Trypsin reduces the barrier properties of sputum and can be used for the combined therapy of respiratory viral infections [13,14].

Chymotrypsin (EC 3.4.21.1) is also a serine endopeptidase secreted by the pancreas. Chymotrypsin precursor—chymotrypsinogen is activated by trypsin. Chymotrypsin preferentially cleaves peptide bonds where the *N*-terminal amino acid is a large hydrophobic amino acid (tyrosine, tryptophan and phenylalanine) but can also hydrolyze other amide bonds in peptides at slower rates, particularly those containing leucine and methionine.

Antioxidant enzyme catalase (EC 1.11.1.6) is Mn III containing hemoprotein with four iron-containing heme groups which catalyzes the decomposition of hydrogen peroxide to water and oxygen in almost all aerobic organisms for protection from oxidative damage by reactive oxygen species (ROS). Catalase is also involved in aging, inflammation, apoptosis, ethanol metabolism and cancer. H_2_O_2_ interferes with the production of melanin, the pigment that gives hair its color. Therefore, low levels of catalase may play a role in hair graying. Catalase is used in the food industry for removing hydrogen peroxide from milk before cheese production and in food wrappers to prevent food oxidizing; in the textile industry to make peroxide-free fabrics; and in contact lens hygiene. The catalase test is important for identifying bacterial species.

Horseradish peroxidase (HRP) (EC 1.11.1.7) is a redox glycoenzyme with a ferroprotoporphyrin group at the active site, which is naturally found in horseradish roots. It catalyzes the oxidation of various protons with peroxide or alkylperoxides. Due to its low cost and broad substrate specificity, HRP is widely used in immunodiagnostics for the detection of proteins and nucleic acids. HRP, in combination with the naturally occurring plant growth phytohormone IAA, has demonstrated antitumor activity in vitro and in vivo [11]. It is suitable for removing diluted emulsified oil and dissolved oil from water [15].

Lipases from both animal and bacterial sources catalyze the hydrolysis of fats and are involved in metabolism of dietary triglycerides in most organisms, cell signaling and inflammation. Besides that, lipases are widely used in laundry detergents. Acidic adipocyte triacylglycerol lipase (EC 3.1.1.3) is found in animals, plants, fungi and bacteria. It hydrolyses triglycerides into diglycerides and subsequently into monoglycerides and free fatty acids. The pancreatic enzyme requires a protein cofactor, namely colipase, to counteract the inhibitory effects of bile salts. Blood tests for lipase may be used to diagnose acute pancreatitis and other disorders of the pancreas.

To construct stable protein NP without any cross-linking, a novel method of nanoprecipitation of proteins from their solutions in fluoroalcohol was developed and applied for fabrication of NP from albumin, fibrinogen, immunoglobulin and lysozyme [16]. The current research was aimed at ENP construction, analysis of their conformation stability, enzyme activity and possible toxicity for tissue cultures.

## 2. Results

### 2.1. Structures of ENPs

ENPs were fabricated from fresh solutions of 7 enzymes in fluoroalcohol by nanoprecipitation [16]. Part of the available data of SEM (Figure 1) and AFM (Figure 2 and Figure 3) showed a broad size range of 50–300 nm and irregular shapes with a polydispersity index of 0.1–0.3 and their gradual growth during ENP storage.

Along with the microscopy data (Figure 1, Figure 2 and Figure 3), dynamic light scattering (DLS) revealed original average diameters of ENP of irregular shapes near 100 nm (Figure 4 and Appendix A) similar to sizes of protein NP from BSA, fibrinogen and lysozyme [16]. Without any cross-linking, all the studied ENP remained stable during storage at refrigerator up to 1 year of observations (Figure 4 and Appendix A). Even for proteolytic enzymes trypsin and chymotrypsin free protein molecules and any reduction of sizes of ENP were not observed (Figure 2, Figure 3 and Appendix A), perhaps because of low temperature and water solutions for long-term storage that are inconvenient for protein digestions. On the contrary, aggregation or agglomeration of the original ENP resulted in nanostructures of bigger submicron sizes and microparticles (Figure 2, Figure 3, Figure 4 and Appendix A).

### 2.2. Conformational Changes

The intrinsic property of proteins to form α-helices and β-sheets leads to a complex phase behavior in which proteins can assemble into various types of aggregates, including crystals, liquid-like phases of unfolded or natively folded proteins, amyloid fibrils and NP [17]. The secondary structures of enzymes are important for their catalytic properties.

Conformational analysis of 7 ENP in comparison with fresh solutions of the same enzymes in water and fluoroalcohol HFIP was carried out by circular dichroism (CD) spectroscopy (Appendix A). Changes in the ellipticity of the enzymes in the original fresh solutions in water and in ENP are shown in Table 1.

CD spectra of trypsin, chymotrypsin and especially catalase in water solutions and in ENP were similar (Appendix A, Table 1) suggesting partial stability of their secondary structures. However, deconvolution analysis revealed a significant decrease in the content of α-helixes in DNase I and HRP NP (Table 1) despite the previous hypothesis of stability of α-helixes in fluoroalcohol solutions [18,19] used for the fabrication of NP. Heating at 58 °C for rapid evaporation of two alcohols destroys secondary structures of the enzymes without complete recovery in the resulting ENP. One should note the absence of a direct correlation between the secondary structures of proteins and their enzymatic activities.

### 2.3. Enzymatic Activity of NP

Despite the conformational changes of the studied enzymes (Appendix A, Table 1), the ENP retained activity toward conventional substrates.

Nuclease activity of DNase and RNase NP was analyzed using electrophoresis in agarose gel with ethidium bromide staining (Figure 5A) and reverse transcription with subsequent real-time PCR, as previously described [20] (Figure 5B, Appendix A). The negative results of reverse transcription with subsequent real-time PCR (Figure 5B, Appendix A) in the presence of RNase NP, DNase NP or trypsin NP at a high concentration of 1 mg/mL suggested exhaustive hydrolysis of RNA, cDNA or Taq DNA-dependent DNA polymerase, respectively. Based on the calibration dependence between the fluorescence threshold cycles (Ct) (Appendix A) and quantities of genome-equivalents in reaction mixtures, as well as by using the Lukyanov-Matz equation N = 2^40−n^, where n is threshold cycle (Ct), and N is the original number of genome-equivalents in the PCR mixture, one can estimate that the treatment with 0.1 mg/mL DNase NP caused cDNA digestion with only 0.003% of the original cDNA left. In the presence of DNase NP at the reduced concentration 0.001 mg/mL nearly 30% of the original cDNA remained (Appendix A). Taking into consideration the high sensitivity of the PCR and non-linear dependence between Ct, genome-equivalents in the reaction mixtures and DNase I activity, direct calculations of the nuclease activity in water or HFIP solutions and in the DNase NP were hardly possible.

Both trypsin (Appendix A) and chymotrypsin NP could digest proteins at +37 °C at acid pH in the presence of 40 μM HCl but remained relatively stable during long-term storage of the protease NP in water at +4 °C (Figure 2, Figure 3 and Appendix A). Multiple products of non-exhaustive protease hydrolysis (Appendix A) did not permit the quantitative comparison of trypsin and chymotrypsin activities in fresh solutions in water and fluoroalcohol, as well as in the protease NP, during long-term storage.

The enzymatic activity of the ENP, consisting of antioxidant enzymes, significantly differed in the colorimetric reactions with H_2_O_2_ in the presence of TMB (Figure 6). HRP remained active in water and HFIP solutions as well as in the ENP even at low concentrations in a broad range 0.001 ÷ 1 mg/mL in enzymatic assay, whereas the fresh solution of the catalase in fluoroalcohol HFIP and catalase NP in the same assay demonstrated only trace activity at concentration 1 mg/mL but not after storage for 1 month (Figure 6).

Direct comparison of HRP activity (Figure 7A) proved that the enzyme remained active in fresh HFIP solution despite conformational changes (Table 1, Appendix A) and from 6.4 to 59.7% of the original HRP activity of fresh solution in water remained in the corresponding ENP at different concentrations. Despite the similar CD spectra (Appendix A, Table 1), the catalase activity in the ENP was in a range of 1.7–3.8% of the original fresh enzyme solution in water (Figure 7B).

Lipase activity in fresh solutions in water and HFIP, as well as in NP was assayed with 2 substrates—OxiRed and 4-nitrophenol yellow (Figure 8 and Appendix A). Surprisingly, the lipase remained active in the HFIP solution but less active in the NP in water with both substrates. The reduced catalytic activity of lipase in ENP may be caused by heating at 58 °C for ~30 min for NP manufacturing. Noteworthy, the calculations of lipase activity with the different substrates (Figure 8 and Appendix A) differed. According to the OxiRed assay, only 4.3% of the original lipase activity in fresh solution in water remained in the lipase NP (Figure 8A), whereas using 4-nitrophenol yellow revealed 18.6% lipase activity in the ENP (Figure 8B).

### 2.4. Toxicity of ENP for Human Cell Lines

ENP were added to wells of polystyrene plates with 80% confluent monolayers of three adhesive cell lines—Vero, HEp-2 and HT-29. At 1 day posttreatment with DNase, chymotrypsin and catalase NP, both human and Vero cells contained multiple granules (Figure 9), but a part of them remained viable with active mitochondrial enzyme—reductase, as shown by the MTT test, in 2–4 days after the ENP addition (Table 2, Figure 10).

Incubation of the cells with RNase NP for up to 4 days did not cause any visible changes in cellular growth compared to control cells (Figure 9), with a non-significant reduction of MTT test results (Figure 10). In the presence of trypsin and HRP NP slower growth resulted in incomplete cellular monolayers (Figure 9 and Figure 10). Moreover, catalase, chymotrypsin and especially DNase NP caused visible granulation of cells (Figure 9) and evident toxicity according to the MTT test (Figure 10, Table 2).

## 3. Materials and Methods

### 3.1. Materials

Deoxyribonuclease I (DNase) from bovine pancreas, ribonuclease (RNase) A, trypsin, chymotrypsin from bovine pancreas, catalase, horseradish peroxidase (HRP), lypase and 1,1,1,3,3,3-Hexafluoro-propan-2-ol (HFIP) of 95–99% purity were purchased from Sigma-Aldrich (St. Louis, MO, USA).

### 3.2. Methods

#### 3.2.1. Protein Nanoprecipitation

ENP were constructed by means of nanoprecipitation according to Morozova et al. [16]. Enzymes were dissolved in 1,1,1,3,3,3-hexafluoroisopropanol (HFIP) until a final concentration of 10 mg/mL. Then, the enzyme solutions in HFIP were added dropwise to 40% ethanol in water (1/10 part of the total volume). Permanent vigorous stirring was necessary to maintain the stability of the three-phase system, including two solvents (water and HFIP) and one anti-solvent (ethanol) of proteins. The mixture was immediately placed at 58 °C (HFIP boiling temperature). To accelerate evaporation of both alcohols, nanoprecipitation was performed under pressure of less than 25 mBar for 30 min. Then water-insoluble particles were pelleted at 14,000× *g* and washed with deionized water 3 times to remove the residual free protein molecules. Microparticles were removed by additional differential centrifugation at 700–1000 g, leaving the ENP in the supernatants.

#### 3.2.2. Scanning Electron Microscopy (SEM)

For SEM, highly oriented pyrolytic graphite (HOPG) (ZYB quality, NT-MDT, Russia) modified with *N*,*N*′-(decane-1,10-diyl)bis(tetraglycineamide) (GM) (Nanotuning, Chernogolovka, Russia) was used as a support according to Barinov et al. [5]. In brief, 10 μL of 10 μg/mL GM solution in double-distilled water was loaded onto a freshly cleaved HOPG surface for 10 s, followed by the removal of the droplet from the surface by a nitrogen flow. Then 1 μL ENP was deposited onto a GM-HOPG surface for 1 min and subsequently rinsed with 100 μL of water followed by immediate removal of the whole droplet by a nitrogen flow. The samples were characterized using a Zeiss Merlin microscope equipped with GEMINI II Electron Optics (Zeiss, Jena, Germany). The SEM parameters were accelerating voltage (1–3 kV) and probe current (30–80 pA). The software “SmartSEM” Version 5.06 was used to analyze SEM images (10 images per sample under different magnifications).

#### 3.2.3. Atomic Force Microscopy (AFM)

Aliquots of 1 μL freshly prepared ENP in water or the same ENP after storage at +4 °C were placed on the freshly cleaved mica surface for 1 min. Then 100 μL water was added to the surface, followed by immediate removal of the whole droplet by a nitrogen flow. ENP were analyzed using a Ntegra Prima (NT-MDT, Moscow, Russia) atomic force microscope. All AFM observations were performed with high-resolution silicon cantilevers with resonance frequencies from 190 to 325 kHz in the attraction regime of intermittent contact mode at a scan rate of 1 Hz. The free amplitude of the cantilever in the air was in the range of 1–10 nm. For standard image processing and presentation, FemtoScan Online (Advanced Technologies Center, Moscow, Russia) was used for height analysis—SPM Image Magic (http://spm-image-magic.software.informer.com, accessed on 1 September 2021).

#### 3.2.4. Dynamic Light Scattering (DLS)

The ENP hydrodynamic radiuses were determined by dynamic light scattering (DLS) using NANO-flex 180° (Microtrac, St. Petersburg, Russia). The refraction index (n) of any protein NP was 1.4, n (water) = 1.3 and n (HFIP) = 1.27 [10]. All measurements were performed at room temperature in deionized water.

#### 3.2.5. Ultraviolet (UV) Spectroscopy

The UV absorption spectra of the enzyme solutions in water and corresponding ENP were determined using a NanoDrop 2000c UV-Vis spectrophotometer (Thermo Scientific, Waltham, MA, USA).

#### 3.2.6. Circular Dichroism Spectroscopy

To probe protein conformational changes, circular dichroism (CD) spectra of 7 enzyme solutions in water, HFIP and corresponding ENP suspensions were obtained by a Chirascan spectrophotometer (Applied Photophysics, Leatherhead, UK). For quantitative analysis of CD spectra, CDNN 2.1 (bestsel.elte.hu Gerald. Bohm, 1997, CD Spectra Deconvolution, Delphi, Halle) software was used.

#### 3.2.7. DNase Activity Assay

Double-stranded circular plasmid DNA and cDNA of β-coronavirus SARS-CoV-2 were treated with DNase I and DNase NP at 37 °C for 30 min and then analyzed in real-time PCR [20]. PCR products were treated with DNase NP at 37 °C for 30 min and analyzed by electrophoresis in 2% agarose gel in Tris-borate-EDTA (TBE) buffer with ethidium bromide staining.

#### 3.2.8. RNase Activity Assay

The isolated total RNA (5 µg) diluted in 5 µL sterile deionized water was incubated with RNase A and RNase NP at 37 °C for 30 min. Then, RNA was assayed in reverse transcription with subsequent real-time PCR [20] and by electrophoresis in 2% agarose gels in TBE buffer with 0.1% SDS and ethidium bromide staining.

#### 3.2.9. Trypsin and Chymotrypsin Activity Assay

Trypsin and chymotrypsin activity were estimated using SDS-PAAG electrophoresis [21].

#### 3.2.10. Catalase and HRP Activity Assay

Catalase and HRP activity were assayed using 3,3′,5,5′-tetramethylbenzidine (TMB) with hydrogen peroxide. The optical density was measured at 450 nm.

#### 3.2.11. Lipase Activity Assay

Lipase activity of NP was analyzed using the lipase activity assay kit (ab102524, Abcam, Waltham, MA, USA) with 2 substrates—4-nitrophenol yellow with spectrophotometric measurements at 410 nm and OxiRed dye by measurements of optical densities at 570 nm according to the manufacturer’s instructions.

#### 3.2.12. MTT Test

Green monkey kidney Vero cells, human larynx carcinoma HEp-2 cells, and human colorectal adenocarcinoma epithelial HT-29 cells were obtained from the Russian State Tissue Culture Collection (National Research Center of Epidemiology and Microbiology, Moscow, Russia) and grown in culture medium 199 (https://paneco-ltd.ru/catalog/pitatelnaya-sreda-199 accessed on 1 March 2021) supplemented with 8% fetal bovine serum (HyClone, Thermo Scientific, USA) in the presence of 0.6 mg/mL linkomycin and 2.5 µg/mL amphotericin-B at 37 °C and 5% CO_2_ until ~80% confluent monolayers.

Toxicity of 7 types of ENP was analyzed for the three cell lines by means of MTT test in dynamics. In brief, sterile MTT solution (5 mg/mL) in PBS was added to each well and incubated for 4 h at 37 °C. After removal of the culture medium and washes, 100 μL of DMSO was pipetted into each well and carefully mixed to solubilize the crystal formazan. The absorbance was measured at 570 nm.

## 4. Discussion

Protein NP can be constructed by different methods of two categories: emulsification and precipitation. Nanoprecipitation is based on the gradual reduction of the solvent in which the main constituent of the NP is dissolved. Depending on the approach used to reduce the solvent quality, nanoprecipitation can be classified into different subtypes, such as non-solvent precipitation, desolvation, coacervation, salting out and albumin-bound technology [22]. Our method of protein nanoprecipitation is based on non-solvent precipitation [16]. Non-solvent precipitation includes three steps: generation of supersaturation, nucleation and growth of NP. Proteins were dissolved in fluoroalcohol as a solvent, while 40% ethanol was used as a poor solvent for proteins. By vigorous mixing of alcohols with water, protein supersaturation permits the formation of protein nuclei with subsequent condensation of free protein molecules around the nuclei, thus creating protein NP [16,22]. This method does not require the use of toxic cross-linkers, such as glutaraldehyde. The intrinsic property of proteins to form structural motifs, such as α-helices and β-sheets leads to aggregation in NP [17]. Protein nanoprecipitation was previously developed for albumin. The protein NP were shown to retain their abilities to bind with ligands and specific antibodies [16]. Enzymatic activity was earlier shown for lysozyme NP capable of destroying bacterial cell walls [16]. It corresponds to the metastability of alcohol-denatured lysozyme [23]. One possible reason is known to be the stability of α-helixes of the lysozyme active center in fluoroalcohol solution. The active site of lysozyme consists of a deep crevice, which divides the protein into two domains linked by an alpha helix [23]. The structure of the lysozyme is consistent under a variety of conditions. Lysozyme appeared to be thermally stable and active at pH 6–9. To our knowledge, the structures and functions of the ENP consisting of nucleases, proteases, lipase and antioxidant enzymes have not been previously studied.

SEM, AFM and DLS demonstrated ENP of irregular shapes with broad overlapping sizes ranging from 50 to 300 nm, similar to the sizes of protein NP from BSA, fibrinogen and lysozyme, without strict correlation between original enzyme molecular weight or protein concentrations and ENP diameters (Figure 1, Figure 2, Figure 3 and Appendix A) [16]. The smallest ENP were fabricated from lysozyme (MW 14.7 kDa, NP sizes 10–40 nm) [16] and chymotrypsin (MW 499.5 kDa, NP sizes 50–100 nm) (Figure 1). Usually, NP sizes range from a few nanometers to several hundreds nm, which enables endocytosis-mediated cellular uptake and an enhanced permeability and retention (EPR) effect in tumor tissues [3].

CD spectra of the enzymes in HFIP solutions (Appendix A) confirmed the stability of α-helixes in fluoroalcohols [18,19]. It is noteworthy that CD spectra (Appendix A) and their deconvolution analysis (Table 1) differed for the enzymes in HFIP used for the nanoprecipitation of the proteins and in the resulting ENP. Trypsin, chymotrypsin and catalase NP had similar secondary structures in their water solutions and in the corresponding ENP (Table 1, Appendix A), but for DNase NP, HRP NP (Table 1) and especially for lipase NP, a significant decrease in content of α-helixes was found despite known stability of α-helixes in HFIP solution [18,19]. RNase A, with a small molecular weight of 13.7 kDa, is composed of three α-helixes and seven β-strands arranged in two “lobes”. Conformations of RNase A in water solutions and NP were shown to differ significantly (Table 1, Appendix A). Nevertheless, due to the well-known remarkable stability of RNase active center RNase, NP produced by nanoprecipitation did not lose their catalytic properties (Figure 5B, Appendix A).

Nuclease activities of DNase and RNase NP were shown by means of gel electrophoresis of DNA (Figure 5A) and RNA before and after treatment with the enzymes in water and in NP, followed by ethidium bromide staining and by the most sensitive detection method—reverse transcription with real-time PCR (Figure 5B, Appendix A). The last approach confirmed the exhaustive hydrolysis of cDNA and RNA in the presence of nuclease NP.

Proteolytic hydrolysis after addition of the trypsin NP (Appendix A) and chymotrypsin NP despite long-term stability of both protease ENP (Figure 2, Figure 3 and Appendix A) could be explained by storage of the ENP at cold temperatures (+4 °C) in water, thus preventing protease autocleavage.

CD spectra of catalase in water, HFIP and NP coincided well (Appendix A), but catalase activity in HFIP solution and especially in the ENP with hydrogen peroxide and TMB was weak compared to HRP solution in HFIP and HRP NP (Figure 6 and Figure 7). Catalase is a tetramer of four polypeptide chains, each over 500 amino acids long with four iron-containing heme groups which permit the destruction of hydrogen peroxide. The core of each subunit is generated by an eight-stranded antiparallel β-barrel (β1-8), with nearest neighbor connectivity capped by β-barrel loops on one side and α9 loops on the other. A helical domain at one face of the β-barrel is composed of four C-terminal helices (α16, α17, α18, and α19) and four helices derived from residues between β4 and β5 (α4, α5, α6, and α7). The optimum pH for human catalase is approximately 7, and the rate of reaction does not change appreciably between pH 6.8 and 7.5. Catalase has one of the highest turnover numbers of all enzymes; one catalase molecule can convert millions of hydrogen peroxide molecules to water and oxygen each second. Catalase solution in HFIP remained colored, suggesting the presence of hemoprotein, but pellets of the catalase NP were white. The stable secondary structure of the catalase in fluoroalcohol HFIP and in the catalase NP (Appendix A, Table 1) did not provide significant catalytic activity (Figure 6 and Figure 7). Possible reasons for the limited activity and low stability of the catalase NP are the aggregation in microparticles of various shapes and sizes (Figure 1), the loss of the iron-containing heme groups during the catalase NP precipitation and probable unavailability of the substrate-binding and catalytic centers.

HRP is a large α-helical brown protein with a molecular weight of 40 kDa that binds heme as a redox cofactor. It consists of a colorless protein (apo-enzyme) combined with an iron-porphyrin. HRP is stable up to 69 °C and from pH 4.5 to 12. By adding the sugars and zinc ions, the peroxidase retains an activity until ~75 °C. Due to the high stability of the HRP, the corresponding ENP remained active after storage at +4 °C for 1 year despite the conformational rearrangements (Appendix A, Table 1). From a biomedical point of view, HRP presents numerous advantageous features, namely, biocompatibility, high stability, high catalytic activity at neutral pH, and the possibility of conjugation to antibodies and NP.

Lipase is a highly soluble enzyme in water and acts on the surface of oil droplets. Access to the active site is controlled by the opening of a lid, which, when closed, hides the hydrophobic surface surrounding the active site. The lid opens when the enzyme contacts an oil–water interface (interfacial activation). The lipase activity with two different specific substrates remained low even in the fresh solution in HFIP (Figure 8A,B) and in the resulting lipase NP was in a range 4.3–18.6% from the original fresh lipase solution in water. Consequently, the method of protein nanoprecipitation is not convenient for the nanotechnological construction of the lipase NP.

Bioactive enzymes interact with their specific substrates and other ingredients involved in cellular metabolism under mild conditions (temperature near 36.7 °C, aqueous media, pH 7.5–8.0) [3]. In the absence of specific receptors, clathrin-caveolar independent endocytosis is involved. NP internalization is based on the endocytic pathway through which the particles remain trapped in endosomes and lysosomes. Unspecific endocytosis-mediated cellular uptake of protein NP includes early and late endosomes and lysosomes, where proteolytic hydrolysis of foreign proteins forms oligopeptides necessary for antigen presentation and immune response induction [24]. All the protection mechanisms against nanomaterials of different origin should be taken into consideration in order to improve endosomal escape, to trigger either intracellular or extracellular controlled drug release [3] and to optimize targeted delivery, relative stability and biodistribution of novel biomaterials such as the ENP. Currently, tissue distribution, cellular internalization, intracellular trafficking, and drug release for a wide variety of already developed bionanomaterials, with some of them under clinical trials, can be rationally controlled [2]. Concerns about their biosafety should be a key issue [3].

Besides the required evaluation of ENP biocompatibility with intracellular enzymes within physiologically acceptable concentration limits on the basis of the MTT test (Figure 10, Table 2) and sulforhodamine B test in order to estimate long-term cell killing [8], biodegradability of the nanocarriers has to be taken into consideration [2]. Since enzymatic activities vary among different patients and diseases, and many enzymes may share similar cleavage sites for compensation of possible deficiency, then the correlation between the certain enzyme and pathology should be carefully explored before clinical implementations.

## 5. Conclusions

The evident advantages of the ENP from nucleases, proteases, lipase and anti-oxidant enzymes were their stability without any cross-linking for 1 year at +4 °C, minor conformational changes in comparison with the original enzymes and the detectable catalytic activities with specific substrates. Cellular uptake of the bioactive ENP did not cause significant toxic effects, probably due to cellular defense based on the entrapment and biodegradation of foreign nanomaterials in lysosomes. Among 7 studied enzymes, DNase NP appeared to be the most toxic for eukaryotic cells. Further research for targeted extracellular or intracellular delivery, endosomal escape and controlled layer-by-layer release of the free active enzymes is required.

## Figures and Tables

**Figure 1 ijms-24-03043-f001:**
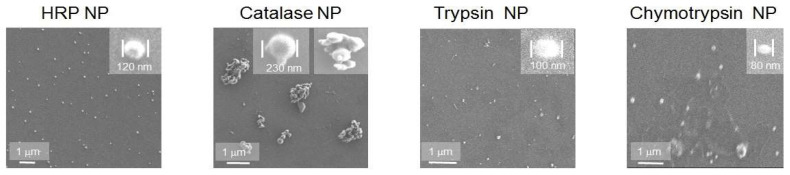
Scanning electron microscopy (SEM) images of ENP consisting of horseradish peroxidase (HRP), catalase, trypsin and chymotrypsin. Scale bar 1 μm and a single ENP with the corresponding diameter are shown for each image.

**Figure 2 ijms-24-03043-f002:**
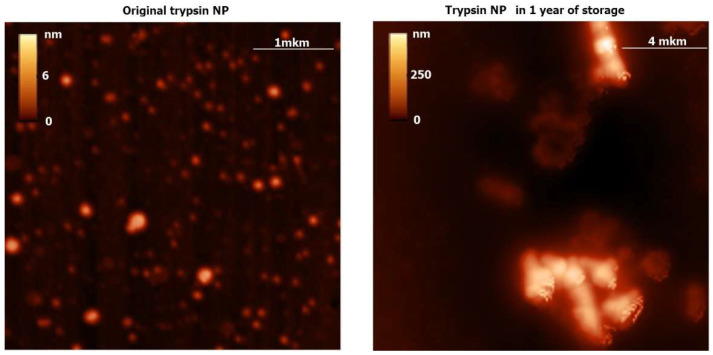
Atomic force microscopy (AFM) images of the original trypsin NP immediately after fabrication and in 1 year of storage at +4 °C.

**Figure 3 ijms-24-03043-f003:**
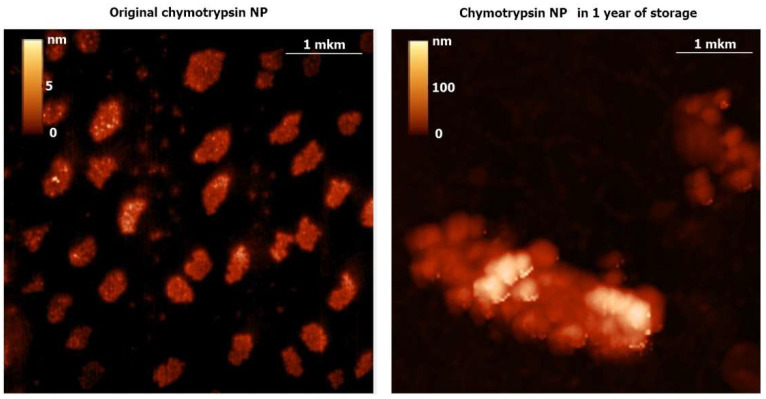
AFM data of the original chymotrypsin NP immediately after fabrication and in 1 year of storage at +4 °C.

**Figure 4 ijms-24-03043-f004:**
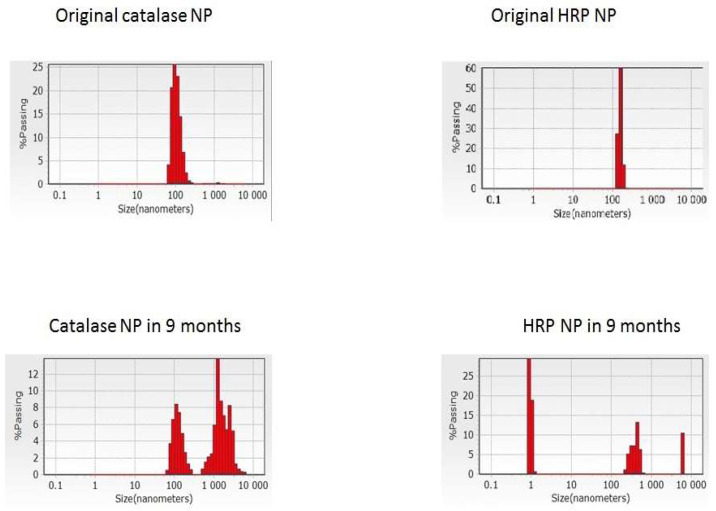
Dynamic light scattering (DLS) measurements of the catalase and HRP NP sizes before and after storage in water for 9 months.

**Figure 5 ijms-24-03043-f005:**
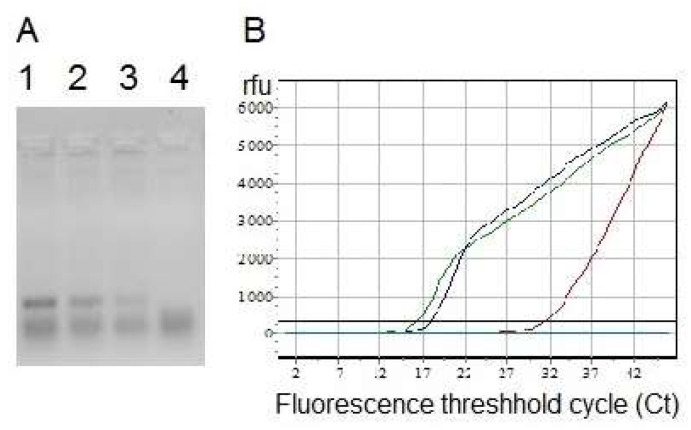
Analysis of the nuclease activities of ENP. (**A**) Electropherogram of PCR product of 153 bp long before (lane 1) and after treatment with DNase NP 0.001 mg/mL (lane 2), 0.01 mg/mL (lane 3) and 1 mg/mL (lane 4) in 2% agarose gel 1× TBE buffer with ethidium bromide staining. (**B**) Data of reverse transcription with real-time PCR for β-coronavirus SARS-CoV-2 RNA from COVID-19 patient blood leukocytes in the presence of the ENP consisting of DNase I, RNase A or trypsin. Calculation of the fluorescence threshold cycles (Ct) for each well is shown in Appendix A.

**Figure 6 ijms-24-03043-f006:**
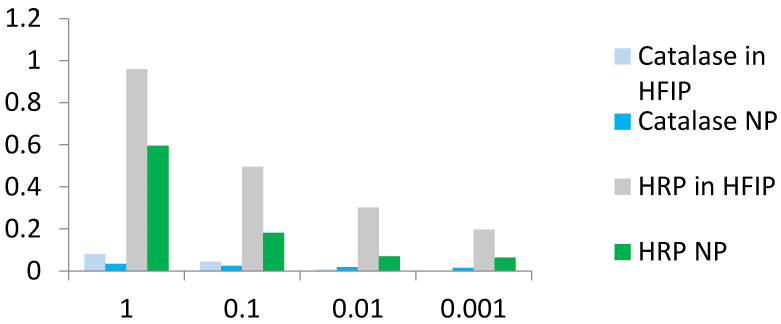
Comparison of enzymatic activity of two antioxidant enzymes in the fluoroalcohol HFIP solution and ENP. Optical density at 450 nm of TMB solution in the presence of H_2_O_2_ and catalase in HFIP solution, catalase NP, HRP in HFIP solution or HRP NP.

**Figure 7 ijms-24-03043-f007:**
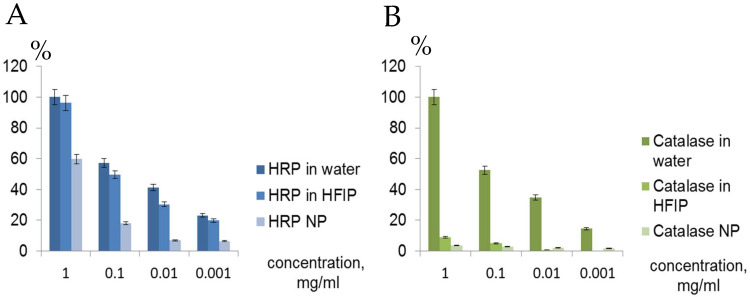
Comparison of enzymatic activity of HRP (**A**) and catalase (**B**) in solutions in water and fluoroalcohol, as well as in ENP.

**Figure 8 ijms-24-03043-f008:**
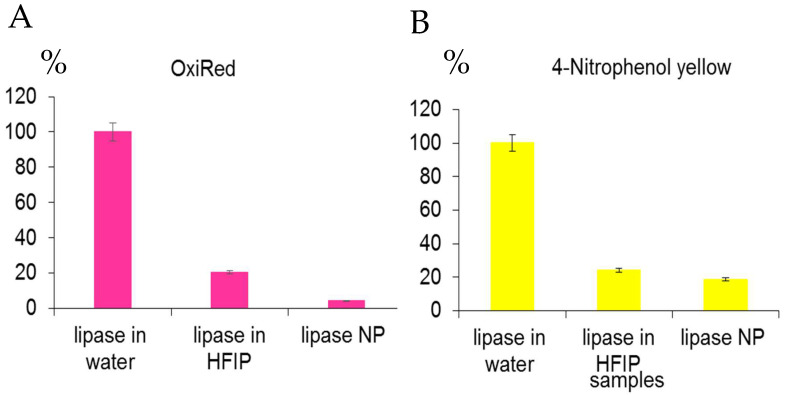
Comparison of lipase activity in water, HFIP solutions and in the ENP using two substrates—OxiRed dye by measurements of optical densities at 570 nm (**A**) and 4-nitrophenol yellow with spectrophotometric measurements at 410 nm (**B**).

**Figure 9 ijms-24-03043-f009:**
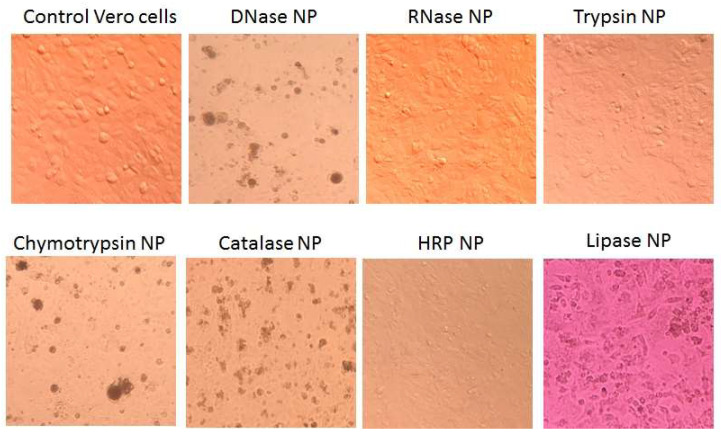
Viability of Vero cells in the presence of 0.1 mg/mL ENP. Optical microscopy images (×400).

**Figure 10 ijms-24-03043-f010:**
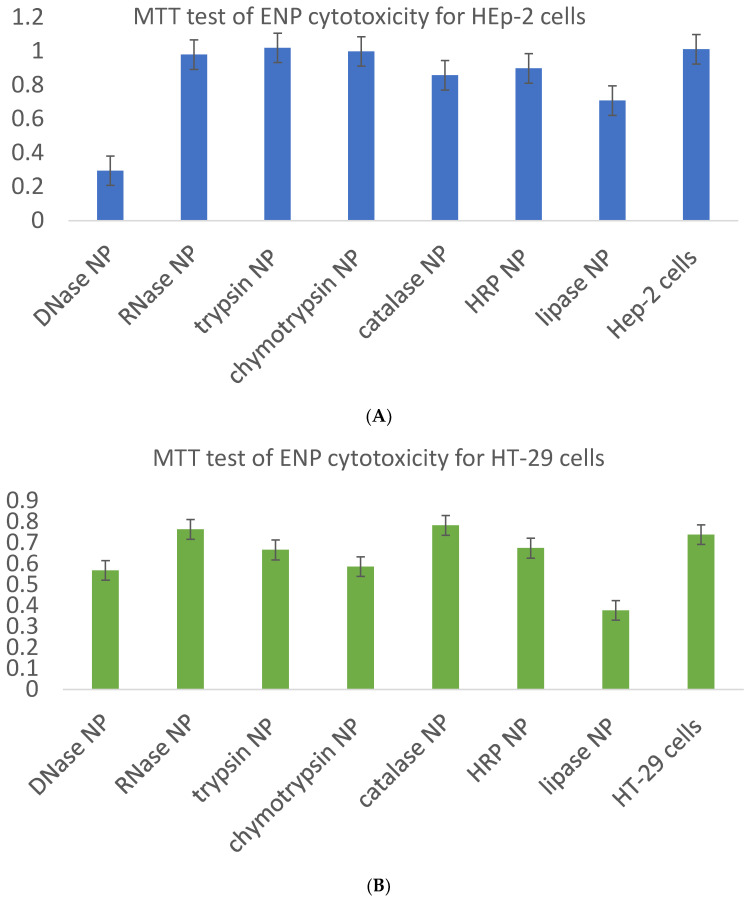
Cytotoxicity of the ENP for HEp-2, HT-29 and Vero cells in the presence of ENP at 3 days posttreatment. Part (**A**) corresponds to HEp-2 cells, part (**B**)—HT-29 cells, part (**C**)—Vero cells.

**Table 1 ijms-24-03043-t001:** Deconvolution analysis of the circular dichroism (CD) spectra of the enzymes in water solutions (E) and nanoparticles (ENPs).

Secondary Structures	DNase I		RNase A		Trypsin		Chymotrypsin		Catalase		Horse Radish Peroxidase	
Structures (%)	E	ENP	E	ENP	E	ENP	E	ENP	E	ENP	E	ENP
Helix	24.1	4	0	0	7.7	13.6	5.7	4.1	0.9	0.5	77.2	3.7
Antiparallel	11	26.5	44.6	36.3	21.1	26.6	22.5	26.8	32.5	33.7	0	27.1
Parallel		10.3	0	31.7	11.5	0	11.5	8.6	9.7	9.8	0	11.3
Beta-Turn	13.9	15.3	13.6	0	15.1	12.8	15.4	16.1	15	14.1	0.1	15.3
Random coil	39.1	43.8	41.8	31.9	44.6	47	44.9	44.4	41.9	41.9	22.7	42.6

**Table 2 ijms-24-03043-t002:** Data of MTT test for HEp-2, HT-29 and Vero cells in the presence of ENP at 3 days posttreatment. Absorbance at 570 nm (optical units (o.u.)).

ENP	HEp-2	HT-29	Vero
DNase NP	0.295	0.569	0.149
RNase NP	0.980	0.764	0.545
trypsin NP	1.020	0.666	0.670
chymotrypsin NP	0.999	0.587	0.649
catalase NP	0.858	0.783	0.763
horseradish peroxidase NP	0.899	0.378	0.712
lipase NP	0.709	0.378	0.427
control cells	1.012	0.739	0.694

## Data Availability

All experimental data will be available on request.

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
