# Peer review of "Stable Enzymatic Nanoparticles from Nucleases, Proteases, Lipase and Antioxidant Proteins with Substrate-Binding and Catalytic Properties"

_ijms, 2023, doi:10.3390/ijms24033043_

Round 1
Reviewer 1 Report
Reviewer comments: The manuscript entitled “Structures and functions of enzymatic nanoparticles” by Morozova et al provides information about the behavior of 7 different enzymatic nanoparticles towards 3 different types of cell lines. Their synthesis, characterization, and testing was studied. However, the following comments needs to addressed before it can be accepted for the final publication.
11. The title of the manuscript needs to be revised; it gives a general ideal when anyone reads it. The title looks more like a review article and is distracting from the original work. Therefore, the authors have to be more specific and have to mention in the title what the work actually describes about?
22. The keywords also need to be revised. They need to identify and include the right technical words from their own work rather than giving some instrumental analysis. Having instrumental analysis the work becomes general and the authors need to identify like in what sense their work is so special?
33. The first sentence of last paragraph in Introduction section (Nanoprecipitation of proteins from their solutions in fluoroalcohol was used for fabrication of NP able to bind with specific ligands) is not clear, rather confusing, please check and modify it.
44. The authors prepared 7 different types of ENPs, but they provided the SEM, AFM images of 4 types of ENPs only, why is like that? What about the missing 3 ENPs? Please include.
55. In general for the cytotoxicity studies, it is a common trend that the data needs to be presented in percentage of cell viability or cell death. However, the authors presented in the real optical optical absorbance values as compared with the control samples. In this way, it becomes very difficult to follow the trend. Therefore, the authors need to change the optical absorbance values into the percentage values with regards to the control measurements.
66. Similar to cytoxicity studies, the values to be transformed into percentage or fold change to the other assays too like Enzymatic activity (Figure 6), lipase activity (Figure S5), etc
77. There is no statistical analysis for various assays like Nuclease and protease activities, enzymatic activity, lipase activity, cell viability studies etc. The authors need to analyze the data statistically and has to confirm whether the data is statistically significant or not. The common tests are ANOVA, Student’s T- test etc.
88. I see that the Supplementary information also included in the main manuscript, it has to be separated from the original manuscript and to be provided as a separate file. Please check.
99. Since the manuscript deals with the biological activity studies following the conversion of various enzymes into their nanoforms. Therefore, the authors need to confirm by comparing the original activity of enzymes with that of their nanoforms (after conversion). I don’t see such comparison for any of the enzymes that they tested. Without these studies it is difficult to come up with a conclusion that the conversion of enzymes (macro) into their respective nanoforms is really activating/enhancing the biological pathways.

Author Response
Reviewer comments: The manuscript entitled “Structures and functions of enzymatic nanoparticles” by Morozova et al provides information about the behavior of 7 different enzymatic nanoparticles towards 3 different types of cell lines. Their synthesis, characterization, and testing was studied. However, the following comments needs to addressed before it can be accepted for the final publication.
- The title of the manuscript needs to be revised; it gives a general ideal when anyone reads it. The title looks more like a review article and is distracting from the original work. Therefore, the authors have to be more specific and have to mention in the title what the work actually describes about?
Answer:
We were pleased to have an opportunity to revise our manuscript now entitled “Stable enzymatic nanoparticles from nucleases, proteases, lipase and antioxidant proteins with substrate-binding and catalytic properties”. In the revised manuscript, we hope to solve all the issues raised. Additional information marked in blue color was included in the changed sections “Introduction” and “Discussion” to convince that our research indeed differed from previously published papers and to highlight its novelty.
The title of the manuscript was changed to the following more specific words corresponding to the content.
“Stable enzymatic nanoparticles from nucleases, proteases, lipase and antioxidant proteins with substrate-binding and catalytic properties”.
- The keywords also need to be revised. They need to identify and include the right technical words from their own work rather than giving some instrumental analysis. Having instrumental analysis the work becomes general and the authors need to identify like in what sense their work is so special?
Answer:
The keywords had been also changed in accordance with the new title of the revised manuscript and the recommendation.
Keywords: DNase I, RNase A, trypsin, chymotrypsin, catalase, horseradish peroxidase, lipase, nanoprecipitation of proteins, enzymatic nanoparticles, broad size ranges, stability of the enzymes and aggregation of nanoparticles during storage, cytotoxicity of bioactive enzymatic nanoparticles.
- The first sentence of last paragraph in Introduction section (Nanoprecipitation of proteins from their solutions in fluoroalcohol was used for fabrication of NP able to bind with specific ligands) is not clear, rather confusing, please check and modify it.
Answer:
The sentence of the last paragraph in Introduction section was replaced to the following.
“To construct stable protein NP with ligand-binding and catalytic activity without any cross-linking the novel method of nanoprecipitation of proteins from their solutions in fluoroalcohol was developed and applied for fabrication of NP from albumin, fibrinogen, immunoglobulins and lysozyme [16]”.
- The authors prepared 7 different types of ENPs, but they provided the SEM, AFM images of 4 types of ENPs only, why is like that? What about the missing 3 ENPs? Please include.
Answer:
Our revised manuscript includes 10 multi-panel figures and 2 tables in the main text as well as 5 large figures and one Table as Supplementary Data. The Supplementary Figure S3 contains CD spectra for all the studied enzymes. It is hardly possible to include all available data for the 7 enzymes from 3 independent experiments with 10 SEM images per each sample and storage controls in dynamics in a single manuscript. Therefore, we had to select part of them. All other experimental data will be available on request.
- In general for the cytotoxicity studies, it is a common trend that the data needs to be presented in percentage of cell viability or cell death. However, the authors presented in the real optical optical absorbance values as compared with the control samples. In this way, it becomes very difficult to follow the trend. Therefore, the authors need to change the optical absorbance values into the percentage values with regards to the control measurements.
Answer:
According to Bjorn Ekwall, 1983 the basal cytotoxicity may be determined by using multiple alternative tests. Among them the most reproducible and reasonable data were based on four following tests:
- sulforhodamine B test to determine total protein concentration;
- ATP content in culture media;
- morphological changes of cells;
- рН
Our results of the two last tests were shown on former Figure 7 with visible granulation of the green monkey kidney Vero cells in the presence of some ENP, detachment of the cells and their monolayer destruction as well as pH changes as various colors of the neutral red dye in cultural medium RPMI 1640. Because of two additional figures on request of other reviewers the numbering of the figures changed and this figure has the number 8.
MTT test is based on the enzymatic activity of the mitochondrial reductase towards 3-(4,5-Dimethyl-2-thiazolyl)-2,5-diphenyl-2H-tetrazolium bromide. But the enzyme may remain active in mitochondria of damaged and even detached cells thus misleading to a wrong conclusion. The former Table 2 demonstrated data of MTT test for 3 types of tissue cultures - HEp-2, HT-29 and Vero cells in the presence of ENP in 3 days posttreatment as absorbance at 570 nm (optical units (o.u.)). As one can see the data do not completely coincide for different cell lines and between alternative cytotoxicity tests (Fig. 9 and 10, Table 2). Therefore, the exact calculations of percentage of cell viability or cell death were hardly possible. For possible visual comparison with control intact eukaryotic cells of 3 types and statistical differences in accordance with the recommendation of other reviewer the new Figure 10, parts A, B, C has been added.
- Similar to cytoxicity studies, the values to be transformed into percentage or fold change to the other assays too like Enzymatic activity (Figure 6), lipase activity (Figure S5), etc
Answer:
Two additional figures 7 and 8 demonstrate the comparison of enzymatic activity of catalase, HRP and lipase, respectively, in water solution, in HFIP solution and in the ENP as percentage from the activity of the corresponding enzyme in freshly prepared water solution with the same protein concentration. But we believe that former Figure 6 corresponding to direct comparison of enzymatic activity of two antioxidant enzymes in fluoroalcohol HFIP solution and in ENP in the same reaction with H2O2 and colored dye TMB and Supplementary Data Figure S5 showing lipase activity in freshly prepared water solution, in HFIP solution and in ENP with two alternative substrates are important as the experimental measurements of optical densities.
- There is no statistical analysis for various assays like Nuclease and protease activities, enzymatic activity, lipase activity, cell viability studies etc. The authors need to analyze the data statistically and has to confirm whether the data is statistically significant or not. The common tests are ANOVA, Student’s T- test etc.
Answer:
Statistical comparison between free enzymes and ENP was performed using Student’s t-test. P values <0.05 were assumed to be significant. The statistical significance was added to the revised manuscript.
- I see that the Supplementary information also included in the main manuscript, it has to be separated from the original manuscript and to be provided as a separate file. Please check.
Answer:
Supplementary data have been included in our first submission for the convenience of reviewers and now were removed from the main manuscript and were submitted in a separate file.
- Since the manuscript deals with the biological activity studies following the conversion of various enzymes into their nanoforms. Therefore, the authors need to confirm by comparing the original activity of enzymes with that of their nanoforms (after conversion). I don’t see such comparison for any of the enzymes that they tested. Without these studies it is difficult to come up with a conclusion that the conversion of enzymes (macro) into their respective nanoforms is really activating/enhancing the biological pathways.
Answer:
Comparison of enzymatic activity was shown for all studied enzymes in fresh water solution, in fluoroalcohol HFIP and in ENP on Figures 5, 6, 7, 8, S4, S5 and in Supplementary Table S1. The statistical significance of the observed differences between free enzymes and ENP was added to the revised manuscript. It was additionally discussed in our revised manuscript.

Reviewer 2 Report
Dear Editor,
I accurately reviewed the article
Manuscript Number: 2085338-v1
Title: Structures and functions of enzymatic nanoparticles
submitted to International Journal of Molecular Sciences.
In this article the authors prepared NP from 7 biomedicine-relevant enzymes including DNase I, RNase A, trypsin, chymotrypsin, catalase, horseradish peroxidase (HRP) and lipase, analysis of their conformation stability and enzymatic activity as well as possible toxicity for eukaryotic cells. Methods: The enzymes were dissolved in fluoroalcohol and mixed with 40% ethanol as anti-solvent with subsequent alcohol evaporation at high temperature and low pressure. NP structures were determined by scanning electron microscopy (SEM), atomic force microscopy (AFM), dynamic light scattering (DLS) and UV-Vis spectroscopy. Enzyme conformations in solutions and in NP were compared using circular dichroism (CD) spectroscopy. Activity of the enzymes was assayed with specific substrates. Cytotoxicity of the enzymatic NP (ENP) was studied by MTT test.
The topic is fascinating, but the authors must solve several critical issues.
Introduction
Authors must specify in detail all acronyms used in the text
The authors should better discuss the difference between the transport of biomolecules or enzymes with nanoparticles and the direct rendering of enzymes into nanoparticles, highlighting the advantages and disadvantages of the different preparations as well as of the nanoparticles themselves. Important jobs in the sector to deal with are for example:
1. Multimodal Enzyme Delivery and Therapy Enabled by Cell Membrane-Coated Metal–Organic Framework Nanoparticles
2. Hydrophilic Gold Nanoparticles as anti-PD-L1 Antibody carriers: Synthesis and Interface Properties; Particle and Particle Systems Characterization 2022, 39, 4 Article number 2100282 DOI 10.1002/ppsc.202100282
3. Enzyme-Responsive Nanoparticles for Anti-tumor Drug Delivery; Front. Chem., 30 July 2020 Sec. Nanoscience https://doi.org/10.3389/fchem.2020.00647
4. Synthesis of functionalized gold nanoparticles capped with 3-mercapto-1-propansulfonate and 1-thioglucose mixed thiols and “in vitro” bioresponse; Colloids and Surfaces B: Biointerfaces 142 (2016) 408-416 doi: 10.1016/j.colsurfb.2016.03.016
5. Enzyme-responsive Nanoparticles for Anticancer Drug Delivery; Current Nanoscience, 2016, 12, 38-46;
6. Nucleobases functionalized quantum dots and gold nanoparticles bioconjugates as a FRET system: synthesis, characterization and potential applications; J. Colloid Interf. Sci. 514 (2018) 479-490
Experimental part
all chemical reagents must be reported in the experimental part, specifying the degree of purity and company.
The preparation must be more detailed in order to allow for a repetition of the expressions by the scientific community.
For example: in how many mL of HFIP is the enzyme cleaved?
For SEM measurements, the number of images and measurements taken and the software used for the image analysis must be specified.
For DLS analyses, the concentration of NPs in the sample analysed, temperature and pH must be specified.
Results and Discussion
The experimental preparation parameter are not discussed, but they could be interesting for the readers.
DLS measurements should report intensity in y.
In addition to showing the peaks, the authors should provide the mean value and the polydispersity of the various samples.
Figures
Supporting figures need not be in the main text
Tables
Table 2 needs to be rechecked
Conclusions
The conclusions are superficial, they should instead highlight the advantages and originality of this material compared to others and future prospects
English requires corrections: many sentences are too long. Moreover there are several typos:
In conclusion, the article is suitable for publication, but only after major revisions.
best regards
Author Response
In this article the authors prepared NP from 7 biomedicine-relevant enzymes including DNase I, RNase A, trypsin, chymotrypsin, catalase, horseradish peroxidase (HRP) and lipase, analysis of their conformation stability and enzymatic activity as well as possible toxicity for eukaryotic cells. Methods: The enzymes were dissolved in fluoroalcohol and mixed with 40% ethanol as anti-solvent with subsequent alcohol evaporation at high temperature and low pressure. NP structures were determined by scanning electron microscopy (SEM), atomic force microscopy (AFM), dynamic light scattering (DLS) and UV-Vis spectroscopy. Enzyme conformations in solutions and in NP were compared using circular dichroism (CD) spectroscopy. Activity of the enzymes was assayed with specific substrates. Cytotoxicity of the enzymatic NP (ENP) was studied by MTT test.
The topic is fascinating, but the authors must solve several critical issues.
Answer:
We thank you for the general appreciation of our research. We have revised the manuscript according to your suggestions.
Introduction
Authors must specify in detail all acronyms used in the text.
Answer:
All acronyms have been described at the first mention in the revised text and in the additional separate section “Abbreviations”.
The authors should better discuss the difference between the transport of biomolecules or enzymes with nanoparticles and the direct rendering of enzymes into nanoparticles, highlighting the advantages and disadvantages of the different preparations as well as of the nanoparticles themselves. Important jobs in the sector to deal with are for example:
- Multimodal Enzyme Delivery and Therapy Enabled by Cell Membrane-Coated Metal–Organic Framework Nanoparticles
- Hydrophilic Gold Nanoparticles as anti-PD-L1 Antibody carriers: Synthesis and Interface Properties; Particle and Particle Systems Characterization 2022, 39, 4 Article number 2100282 DOI 10.1002/ppsc.202100282
- Enzyme-Responsive Nanoparticles for Anti-tumor Drug Delivery; Front. Chem., 30 July 2020 Sec. Nanoscience https://doi.org/10.3389/fchem.2020.00647
- Synthesis of functionalized gold nanoparticles capped with 3-mercapto-1-propansulfonate and 1-thioglucose mixed thiols and “in vitro” bioresponse; Colloids and Surfaces B: Biointerfaces 142 (2016) 408-416 doi: 10.1016/j.colsurfb.2016.03.016
- Enzyme-responsive Nanoparticles for Anticancer Drug Delivery; Current Nanoscience, 2016, 12, 38-46;
- Nucleobases functionalized quantum dots and gold nanoparticles bioconjugates as a FRET system: synthesis, characterization and potential applications; J. Colloid Interf. Sci. 514 (2018) 479-490.
Answer:
Both advantages and disadvantages of different nanomaterials for the transfer of enzymes inside eukaryotic cells were included in our revised manuscript with emphasis in the “Conclusion”. Biocompatible and biodegradable nanoparticles constructed from natural biopolymers seem to be preferable and desirable in comparison with metallic and organic materials. However, immune response with possible immune-dependent sequelae such as allergy and autoimmune diseases cannot be excluded. Besides that unspecific endocytosis-mediated cellular uptake of exogenous nanomaterials results in protein degradation in lyzosomes. Therefore, unstable enzymatic nanoparticles are expensive. All the recommended references have been described in our revised sections “Introduction” and “Discussion”.
Experimental part
all chemical reagents must be reported in the experimental part, specifying the degree of purity and company.
Answer:
Materials have been described in details in our revised manuscript including company and purity degrees.
The preparation must be more detailed in order to allow for a repetition of the expressions by the scientific community.
For example: in how many mL of HFIP is the enzyme cleaved?
Answer:
Each enzyme was dissolved in HFIP until final concentration 10 mg/ml. Total volume depended on the goal of the experiment and the number of necessary repeats. For the physical and chemical analysis 100-200 µl was enough, whereas for cytotoxicity assay – up to 1 ml for multiple repeats and various cell lines.
For SEM measurements, the number of images and measurements taken and the software used for the image analysis must be specified.
Answer:
The software “SmartSEM” Version 5.06 was used to analyze SEM images (10 images per each sample under different magnifications).
For DLS analyses, the concentration of NPs in the sample analysed, temperature and pH must be specified.
Answer:
All measurements were performed at room temperature. ENP were in miliQ deionized water but not in a buffer solution, therefore, pH values might slightly vary and were not exact. Our equipment NANO-flex 180° (Microtrac, USA) does not permit to analyze the concentrations of nanoparticles but the only hydrodynamic radiuses.
Results and Discussion
The experimental preparation parameter are not discussed, but they could be interesting for the readers.
Answer:
Our protocol of the protein nanoprecipitation was earlier optimized and described in details in our previous papers (Morozova et al., 2018; 2020).
- Morozova, O.V., Pavlova, E.R., Bagrov, D. V., Barinov, N.A., Prusakov, K.A., Isaeva, E.I., Podgorsky, V.V., Basmanov, D.V., Klinov, D.V. Protein nanoparticles with ligand-binding and enzymatic activities. International Journal of Nanomedicine 2018, 13, 6637–6646. doi:2147/IJN.S177627.
- Morozova O.V., Isaeva E.I., Klinov D.V. Protein Nanoparticles with Enzymatic and Antigen-binding Activities Induce Th1 Cytokine Gene Expression. Materials Science Forum ISSN: 1662-9752, Vol. 995, pp 109-113. doi:10.4028/www.scientific.net/MSF.995.109 © 2020 Trans Tech Publications, Switzerland.
- Morozova V., Sokolova A.I., Pavlova E.R., Isaeva E.I, Obraztsova E.A., Ivleva E.A., Klinov D.V. Protein nanoparticles: cellular uptake, intracellular distribution, biodegradation and induction of cytokine gene expression Nanomedicine 2020; doi: 10.1016/j.nano.2020.102293; PMID: 32853784.
DLS measurements should report intensity in y.
Answer:
ENP hydrodynamic radiuses were determined by dynamic light scattering (DLS) using NANO-flex 180° (Microtrac, USA). The standard software shows “% passing” on the axis Y. The DLS results using the standard software of NANO-flex 180° (Microtrac, USA) are shown on Figure 4 and Supplementary Data Figure S1. They allow us to compare size distribution of nanoparticles in different samples of ENP during long-term storage.
In addition to showing the peaks, the authors should provide the mean value and the polydispersity of the various samples.
Answer:
Mean values of hydrodynamic radiusis 50-300 nm and polydispersity indexes (PDI) 0.1-0.3 of the studied ENP varied from experiment to experiment and depended on their storage time (Fig. 4, Supplementary data Fig. S1, S2). The information was added in our revised text.
Figures
Supporting figures need not be in the main text
Answer:
Supporting data have been included in our first submission for the convenience of reviewers. Supplementary Data including five figures and one table have been removed from the revised main manuscript and submitted in the separate file.
Tables
Table 2 needs to be rechecked
Answer:
Table 2 containing data of MTT test for HEp-2, HT-29 and Vero cells in the presence of ENP in 3 days posttreatment was carefully checked. New Figure 10 with statistical analysis was added according to the recommendation of other reviewer.
Conclusions
The conclusions are superficial, they should instead highlight the advantages and originality of this material compared to others and future prospects
Answer:
Conclusion was revised. The advantages and disadvantages were added. Possible further development was described.
English requires corrections: many sentences are too long. Moreover there are several typos:
Answer:
Our revised text had been carefully checked and modified. The sentences became shorter and our mistakes were corrected. The significant changes of our revised manuscript are highlighted in blue.
In conclusion, the article is suitable for publication, but only after major revisions.
Answer:
The manuscript was revised according to all the recommendations. In the revised manuscript, we hope to solve all the issues raised.

Reviewer 3 Report
Please find the attachment

Author Response
The manuscript titled “Structures and functions of enzymatic nanoparticles” by Morozova et al. reports the fabrication of enzymatic nanoparticles using seven different enzymes. The authors have included the physicochemical evaluations of the formed aggregates and have also analyzed the conformational changes. In the reviewer’s opinion the manuscript lacks direction and appears to be very randomly oriented. In the reviewer’s opinion the manuscript is to be rejected in this form. Here are the few comments/enquiries/suggestions to back the reviewer’s decision.
- The authors should check the journal guidelines for keywords, keywords should be pertinent and precise.
Answer:
Keywords have been changed together with the new more precise and specific title in accordance with the recommendations of another reviewer.
- The introduction section needs further refinement with adequate citation for example “They are used in many industrial and pharmaceutical areas for both diagnostics and therapy”. “Poor membrane permeability and biological barrier-crossing capacities of free enzymes hamper treatment of diseases”. These statements should be cited with relevant studies.
Answer:
The statements have been confirmed with relevant references. Additional references were added in the revised manuscript according to recommendations of other reviewers.
- The statement “ ENP are the aggregates of enzymes or NP” is very confusing, can an aggregate of NP alone be called as ENP? The authors should rephrase such confusing sentences through out the manuscript.
Answer:
- The intrinsic property of proteins to form structural motifs such as α-helices and β-sheets leads to a complex phase behavior in which proteins can assemble into various types of aggregates including crystals, liquid-like phases of unfolded or natively folded proteins, amyloid fibrils and nanoparticles (Auer and Kashchiev, 2010). We developed the method of the nanoprecipitation of proteins for fabrication of stable protein nanoparticles without any cross-linking (Morozova et al., 2018). Such protein nanoparticles possess the secondary structures similar to original protein in water solutions and retain ligand-binding properties (Morozova et al., 2018). Nanoparticles consisting of enzymes can be called ENP.
Auer, S., Kashchiev, D. Phase diagram of alpha-helical and beta-sheet forming peptides. Phys Rev. Lett. 2010, 104, 168105. DOI:10.1103/PhysRevLett.104.168105.
Morozova, O.V., Pavlova, E.R., Bagrov, D. V., Barinov, N.A., Prusakov, K.A., Isaeva, E.I., Podgorsky, V.V., Basmanov, D.V., Klinov, D.V. Protein nanoparticles with ligand-binding and enzymatic activities. International Journal of Nanomedicine 2018, 13, 6637–6646. DOI:10.2147/IJN.S177627.
The statement was from the relevant reference [current number 4 corresponds to the former reference 2].
Chapman R., Stenzel M.H. All Wrapped up: Stabilization of Enzymes within Single Enzyme Nanoparticles. J Am Chem Soc. 2019;141(7):2754-2769. DOI: 10.1021/jacs.8b10338.
- Authors should carefully evaluate the entire manuscript for grammatical errors for example, line 58 the statement should be “In single enzyme NP, the enzyme……” .
Answer:
We would like to express our gratitude to the reviewer for the attention and so careful reading of our manuscript. Our revised text had been carefully checked and modified. Both typos and grammatical errors had been corrected.
The sentence mentioned above was replaced to the following.
“Enzymes can be stabilized by thin shells, consisting of polymers, prepared either by in situ polymerization or by assembling preformed polymers [4]”.
- The structure of the study looks very incomplete, for few SEM is taken for few AFM is taken, for few DLS study is included.
Answer:
Our revised manuscript contains 10 muti-panels figures and 2 tables (in total, 29 pages). Besides that there are Supplementary Data (5 figures and a table). The only Supplementary Figure S3 includes 7 pages with circular dichroism (CD) spectra of all the studied enzymes in freshly prepared water solution, in HFIP solution necessary for nanoprecipitation and in ENP. All the described experiments were repeated at least 3 times. As to SEM and AFM there are 10 images per each sample under different magnifications. Stability of the ENP was studied in dynamics during storage for 1 year. Therefore, it is hardly possible to include all experimental data that can be available on request.
- What is the rational of taking DLS data for only 3 enzymes after 9 months where as AFM of trypsin and chemotrypsin after 1 year ?
Answer:
DLS measurements are available for all 7 enzymes (Figure 4 for catalase NP and HRP NP immediately after their fabrication and in 9 months of storage at +4°C; Supplementary Data Figure S1 for all ENP after their storage). There are 3 independent series of 7 ENP in dynamics to prove their long-term stability, all available data are described in our revised text but not shown in main figures because of overloaded multi-panel figures. Storage of ENP from active proteolytic enzymes was the most unexpected and surprising finding because of possible protease autocleavage in water solutions. However, trypsin and chemotrypsin NP remained stable after storage in water suspensions at +4°C probably due to low temperature and non-optimal conditions for proteolytic hydrolysis.
- The authors have not clarified what table 2 exactly signifies, moreover, there is no graphic representation of the cytotoxicity evaluation which could provide statistical relevance of the data.
Answer:
Former Table 2 showed experimental data of MTT test for HEp-2, HT-29 and Vero cells in the presence of ENP in 3 days posttreatment. New Figure 10 with statistical analysis was added according to the recommendation of other reviewer. But MTT was not the only test to estimate cytotoxicity of the ENP.
According to Bjorn Ekwall, 1983 the basal cytotoxicity may be determined by using multiple alternative tests. Among them the most reproducible and reasonable data were based on four following tests:
- sulforhodamine B test to determine total protein concentration;
- ATP content in culture media;
- morphological changes of cells;
- рН changes.
Our results of the two last tests were shown on former Figure 7 with evident granulation of the green monkey kidney Vero cells in the presence of some ENP, detachment of the cells and their monolayer destruction as well as pH changes as various colors of the neutral red dye in cultural medium RPMI 1640. Because of two additional figures on request of other reviewers the numbering of figures changed and this figure has the number 8.
MTT test is based on the activity of the mitochondrial reductase towards 3-(4,5-Dimethyl-2-thiazolyl)-2,5-diphenyl-2H-tetrazolium bromide (MTT). But the enzyme may remain active in mitochondria of damaged and even detached cells thus misleading to a wrong conclusion. The former Table 2 demonstrated data of MTT test for 3 types of tissue cultures - HEp-2, HT-29 and Vero cells in the presence of ENP in 3 days posttreatment as absorbance at 570 nm (optical units (o.u.)). As one can see the data do not completely coincide for different cell lines and between alternative cytotoxicity tests (Fig. 9 and 10, Table 2). For possible visual comparison with control intact eukaryotic cells of 3 types and statistical differences in accordance with the recommendation of other reviewer the new Figure 10, parts A, B, C has been added. According to MTT test the results did not differ significantly between control intact cells and those in the presence of the ENP. So slight (if any) cytotoxicity of the bioactive ENP can be caused by their entrapment in the cellular endosomes and lysosomes with subsequent protein degradation as one of the known cellular defence mechanism.

Round 2
Reviewer 1 Report
The authors made modifications to my comments and therefore, I accept this manuscript to publication in its present form.
Reviewer 2 Report
the authors have improved the manuscript which is now ready for publication